# Magnitude and nucleation time of the 2017 Pohang Earthquake point to its predictable artificial triggering

Serge A. Shapiro [1✉], Kwang-Hee Kim [2] & Jin-Han Ree[3]

A damaging Mw5.5 earthquake occurred at Pohang, South Korea, in 2017, after stimulating an enhanced geothermal system by borehole fluid injections. The earthquake was likely triggered by these operations. Current approaches for predicting maximum induced earthquake magnitude ($M_{max}$) consider the volume of the injected fluid as the main controlling factor. However, these approaches are unsuccessful in predicting earthquakes, such as the Pohang one. Here we analyse the case histories of induced earthquakes, and find that $M_{max}$ scales with the logarithm of the elapsed time from the beginning of the fluid injection to the earthquake occurrence. This is also the case for the Pohang Earthquake. Its significant probability was predictable. These results validate an alternative to predicting $M_{max}$. It is to monitor the exceedance probability of an assumed $M_{max}$ in real time by monitoring the seismogenic index, a quantity that characterizes the intensity of the fluid-induced seismicity per unit injected volume.

[1] Earth Science Department, Freie Universität Berlin, Malteserstr. 74-100, 12249 Berlin, Germany. [2] Department of Geological Science, Pusan National University, Busan 46241, Republic of Korea. [3] Department of Earth and Environmental Sciences, Korea University, Seoul 02841, Republic of Korea. ✉email: shapiro@geophysik.fu-berlin.de

The ability to accurately predict the expected maximum possible magnitude, $M_{max}$, of a potential artificially triggered earthquake is a critical question that arises during fluid-related geotechnical operations[1–3], such as water reservoir, artificial lake construction and underground waste- and salt-water disposal[1,4–8], enhanced geothermal system (EGS) development[2,9–12], hydrocarbon production[13,14], $CO_2$ capture, sequestration and underground gas storage[15], hydraulic fracturing of rocks[16,17], and coal, mineral, and ore mining[1].

In November 2017 near the city of Pohang, South Korea, shortly after borehole fluid-injection operations targeted to a stimulation of an EGS, an unexpectedly strong earthquake occurred. The earthquake was likely triggered by the EGS-stimulation operations[2,9–11,18–21], which raised questions regarding the ability to forecast the maximum magnitude of an anthropogenic earthquake.

There is considerable active research[22–27] in developing strategies to forecast $M_{max}$. Popular approaches consider the injected fluid volume, $\Delta V_f$, as the controlling parameter[22,24,25]. Some approaches[22,28,29] implicitly assume that fluid injection is the sole source of strain in the surrounding rocks, and conclude that the seismic moment of the largest induced earthquake (along with the cumulative seismic moment of the induced seismicity), $M_0^{max}$, is limited by a quantity that is proportional to the fluid volume, whereby $M_0^{max} \propto \Delta V_f$. Especially in seismically active regions, this singular role of fluid injection on the local strain may be problematic, because this simple assumption fails to adequately account for tectonic sources of strain and effective stress changes[27].

Another approach is to compute the moment of the largest arrested rupture[25] that was induced by pore-fluid pressure perturbation in the layer around an area of the earthquake fault plane. A comparison of the stress intensity factor (a function of the pore-pressure perturbation) and fracture toughness yields a different power law[25], $M_0^{max} \propto \Delta V_f^{3/2}$. In this power law, the proportionality coefficient[25] depends on the background stress drop, friction, elastic properties and thickness of the pressure-perturbed layer, with the seismic moment of the maximum arrested rupture eventually being controlled by the injection-induced pressure perturbation. Such a perturbation is independent of the existing tectonic stresses at the injection site. Thus, such an approach has a potential for failing in tectonically active regions and areas situated near critically stressed faults. However, one can adjust the proportionality coefficient of the power law $M_0^{max} \propto \Delta V_f^{3/2}$ to the tectonic conditions[25] by relating it to the seismogenic index, $\Sigma$, which quantifies the induced seismicity that may be produced at a given injection site in response to a unit injected volume[30,31] (see 'Methods' section).

A direct application of the seismogenic index model[23,30,31] to the largest earthquake in a time series of fluid-induced seismicity has also provided an estimate[24] of $M_{max}$. However, such an estimate is derived from a statistical formulation of the magnitude–frequency distribution and yields the magnitude of an event whose occurrence probability is nearly 63% (see 'Methods' section). This approach provides no constraints on the potential to trigger larger earthquakes.

Recent studies on the 2017 Pohang Earthquake have emphasised that none of the abovementioned approaches are adequate in describing this earthquake[2,9,10,19], whereas case histories[22,25] have shown that $M_{max}$ is significantly correlated with $\Delta V_f$. However, magnitude scaling with $\Delta V_f$ is a function of the geometry of the processes under consideration, as three[22]- and two[25]-dimensional pore-pressure-diffusion-based approaches have yielded various power-law dependencies of $M_0^{max}(\Delta V_f)$. The duration of the earthquake triggering can be more indicative of the underlying physical process. For example, the time scaling of

normal diffusion is the same in both of the abovementioned cases. Moreover, $\Delta V_f$ is closely related to the elapsed time $\Delta T$ from the beginning of fluid injection to the $M_{max}$ earthquake.

Here, we analyse published case histories and show that $M_{max}$ scales with $\log(\Delta T)$. Furthermore, this magnitude scaling approach makes the artificial triggering nature of the 2017 Pohang Earthquake more noticeable than magnitude scaling with $\Delta V_f$. However, $\Delta T$ is unknown a priori; therefore, the seismogenic response of the surrounding rocks due to fluid injection should be monitored in real-time to address the $M_{max}$ problem. We propose to monitor the worst-case probability of a hypothetical $M_{max}$ earthquake and show that such a strategy could be useful in the cases of Pohang and other damaging induced earthquakes (e.g. the Denver seismicity of 1965–67). We show that a significant probability of the 2017 Pohang Earthquake was predictable.

## Results and discussion

**Maximum magnitudes scale with logarithm of elapsed times.** We used existing compilations[22,24,25,28,29] of published $M_{max}$ data for fluid-injection-induced earthquakes, and included $\Delta T$ data from corresponding and some additional literature sources[5–7,12,18,24,31–58]. In some publications times, $\Delta T$ can be directly found. However, in the other cases, we estimated $\Delta T$ from published plots. We provide $\Delta T$ values up to the second digit in the leading order. This precision is sufficient for our consideration below. The data and corresponding references are given in the Supplementary Data 1 file.

We consider $M_{max}$ as functions of the injected fluid volumes, $\Delta V_f$, and the elapsed times, $\Delta T$, (Fig. 1). The complete data set includes field observations of fluid injections in deep boreholes (data points with Mw > −3), and observations obtained from mine-like (underground laboratories) and laboratory experiments (data points with Mw < −3). The laboratory and mine-like observations are characterised by the presence of free-surface-type boundaries in the vicinity of the injection sources: sample surfaces or mining tunnel walls. The results from these experiments must not be described by the same scalings and/or factors as the field observations in deep boreholes since the latter are better approximated by an infinite continuum. We, therefore, consider the field observations of fluid injections in deep boreholes separately to the mine-like and laboratory observations (Fig. 1d, e). We find that $M_{max}$ is well correlated with both $\Delta V_f$ and $\Delta T$ (Fig. 1). However, magnitude scaling with $\Delta T$ describes $M_{max}$ better by providing a more compact set of data points (Fig. 1d, e). For example, a linear regression of the $\Delta T - M_{max}$ scaling provides significantly better statistics than the equivalent fitting of the $\Delta V_f - M_{max}$ scaling (e.g. a significantly higher coefficient of determination and significantly lower standard deviations of fitting parameters (Supplementary Information file, Eqs. 10 and 11)). Moreover, $\Delta T - M_{max}$ scaling is also significantly more successful in describing the 2017 Pohang Earthquake.

Our findings suggest the following. The growth of the characteristic size, $L$, of an impacted rock domain (stimulated volume) due to fluid injections can be governed by various processes, such as (non-)linear pore-pressure relaxation[23], which sometimes includes a significant portion of the poroelastic coupling[59], aseismic deformation[60,61], and/or a diffusion-driven event interaction[11]. The growth of $L$ can be approximated as a power-law function, $\Delta T^\xi$, where $\xi$ is indicative of the physical process(es) controlling this growth. For example, $\xi = 1$ in the case of a ballistic propagating perturbation, $\xi = 1/2$ in the case of a linear pore-pressure relaxation (a normal pore-pressure diffusion), and $1/3 \leq \xi < 1/2$ in the case of a non-linear diffusion-like process (an anomalous diffusion process) of opening additional 3-D pore

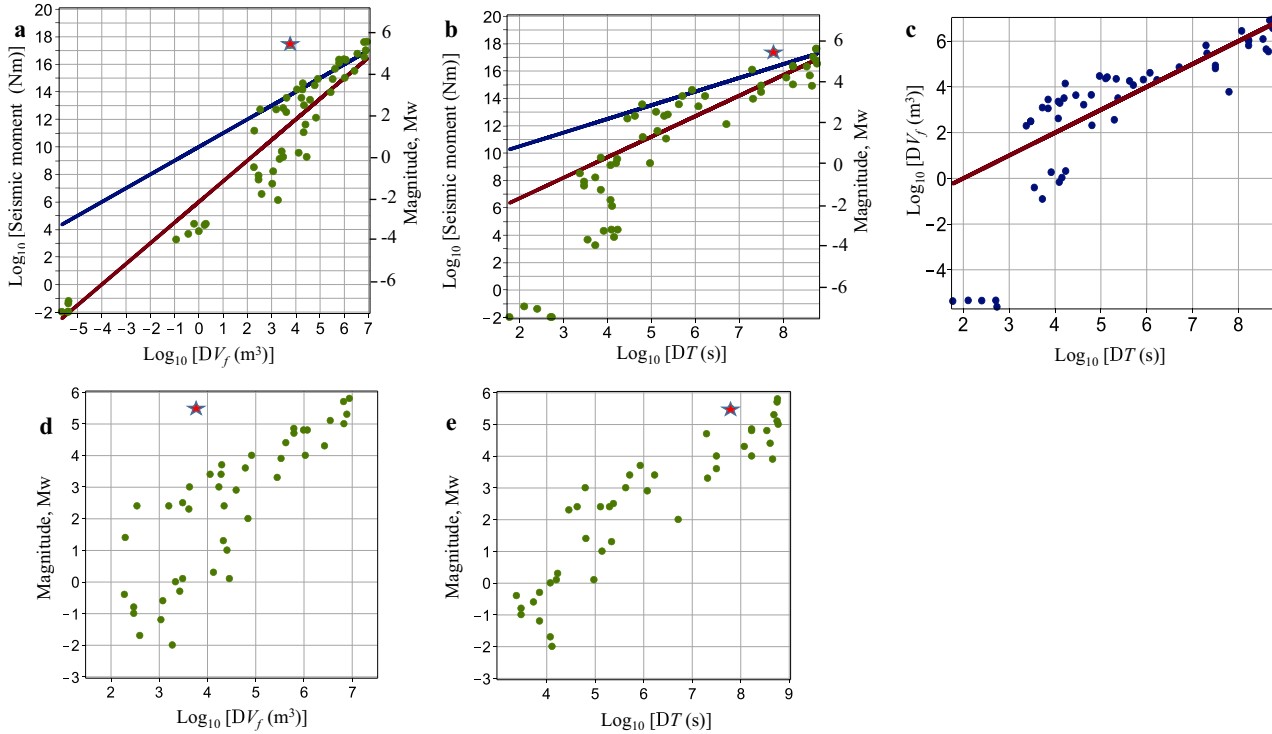

**Fig. 1 $M_{max}$ observations (green points) in fluid-injection environments.** $M_{max}$ vs. **a** injected fluid volume $\Delta V_f$ and **b** elapsed time $\Delta T$ from the beginning of fluid injection until $M_{max}$ occurs. The data include field observations of seismicity induced by injections in deep boreholes[5–7,12, 18, 24,31–56] (data points with Mw > −3), and observations obtained from mine (underground laboratory) and laboratory experiments on small samples[57, 58] (data points with Mw < −3). The straight lines correspond to scaling laws $L \propto \Delta T^\xi$ with $\xi = 1/2$ (red line; normal-diffusion growth) and $\xi = 1/3$ (blue line; it yields a proportionality of the seismic moment to $\Delta V_f$). The Mw5.5 Pohang earthquake[18, 19] is shown by the red stars. **c** $\Delta T$ versus $\Delta V_f$ (the *x*- and *y* coordinates of the blue points) for the data in **a**, **b**; the red line shows a proportionality. **d**, **e** The same as in **a**, **b** but the plots have equal *x*- and *y* axes ranges, respectively, and include only the field observations of injections in deep boreholes.

space[23]. Asymptotically, such power laws apply to both the triggering front and backfront[62] (approximate outer and inner envelopes of the seismically activated stimulated volume). $L$ also continues to grow after the termination of injection[62]. The size $L$ influences the frequency–magnitude statistics of the induced seismicity[23], because the length statistics of triggered rupture surfaces that are related to the stimulated volume (i.e. contacting, intersecting or belonging to it) are controlled by this size[23]. The probability of a triggered earthquake with a large rupture length, $R_{max}$, which corresponds to a given $M_{max}$, increases with increasing $L$. This is also the case for so-called runaway ruptures because increasing the stimulated volume raises the probability that it will contact a large fault[23,25]. This relationship can be approximated as $R_{max} \propto L$ because the contacting probability[23] is a function of $R_{max}/L$. The well-known relationships between the rupture length $R$ of an earthquake and its seismic moment, $M_0 \propto R^3$, and between the seismic moment and the moment magnitude[22,23,24,25], Mw= $2(\log M_0 − 9.1)/3$, then yield $M_0^{max} \propto L^3$ and $M_{max} \approx 2\log_{10} L + const$. The substitution of $L \propto \Delta T^\xi$ into these relationships yields the following scalings: $M_{max} \approx 2\xi\log_{10}\Delta T + const$ and $M_0^{max} \propto \Delta T^{3\xi}$.

We find that the scalings $M_0^{max} \propto \Delta T^{3/2}$ and, respectively, $M_{max} \approx \log_{10}\Delta T + const$ generally describe the field-scale observations (Fig. 1b, e and the linear regression results, Supplementary Information file, Eqs. 10 and 11). These relations correspond to the power law $L \propto \Delta T^{1/2}$, which is typical for the size-time dynamics of rock volumes stimulated by normal pressure diffusion processes. Thus, our observations indicate a statistically dominant role of linear pore-pressure diffusion in triggering $M_{max}$ earthquakes. The following three scenarios[23] can be

relevant for such earthquakes: (i) the rupture surface of the earthquake is contained within the stimulated volume (sometimes strictly such events are called induced earthquakes[18,19,23]); or (ii) the nucleation domain of the rupture surface and its part are contained within the stimulated volume; or finally, (iii) the rupture surface is touched or slightly intersected by the stimulated volume (sometimes events, addressed in the scenarios (ii) and (iii) are called triggered ones[18,19,23]). The cases of ruptures arrested inside or outside of the stimulated volume as well as runaway ruptures are parts of these scenarios. At least in the scenarios (ii) and (iii), magnitudes of triggered earthquakes are very likely controlled by tectonic features of corresponding geological sites rather than by the scale $L$ of the stimulated volume. However, the occurrence probability of such earthquakes depends on this scale, as it follows from our consideration above. The Pohang Earthquake is an example of such an event. Figure 2a–e indicates that it corresponds to the scenario ii. This is also in agreement with the conclusion of the ORAC Committee[18] that the Pohang Earthquake was triggered by an EGS-stimulation impact on the earthquake hypocenter domain. This impact could have various forms, for example, a direct injection-produced pore-pressure perturbation or a combined pore-pressure and stress perturbations[11] of induced earthquakes in the stimulated volume.

The abovementioned magnitude scalings naturally include situations of earthquake triggering after the termination of fluid injection since the law, $L \propto \Delta T^\xi$, also describes the continuing growth of the stimulated rock volume after the termination of injection[62]. These magnitude scalings are therefore more adequate than those with $\Delta V_f$. Large earthquakes have an enhanced occurrence probability at or shortly after the

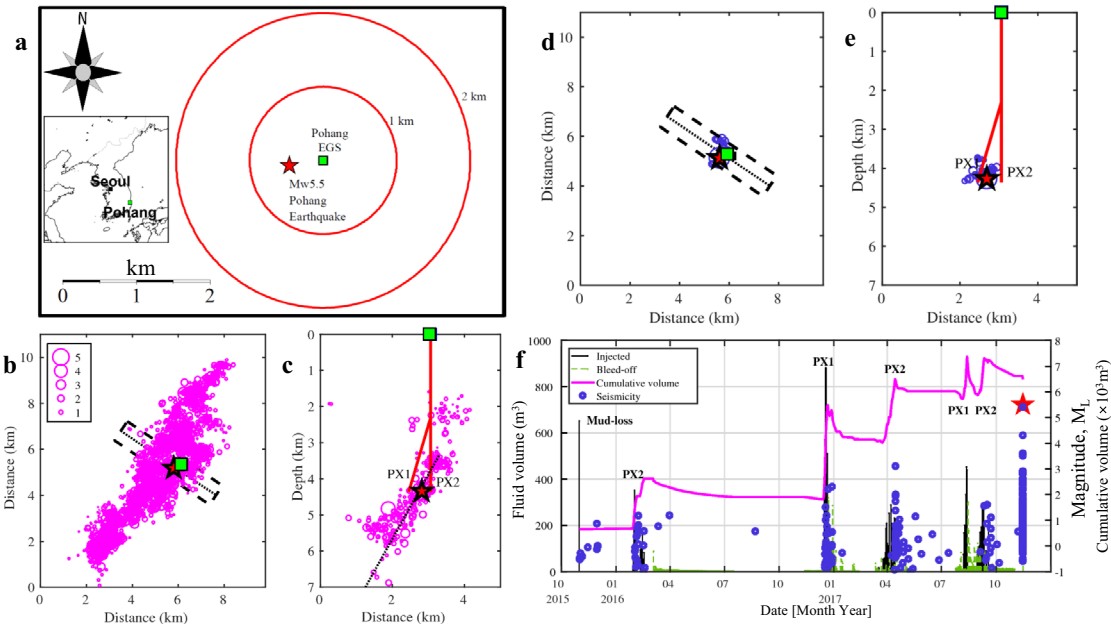

**Fig. 2 Seismicity and fluid injection at the Pohang Geothermal Site[11, 18, 20].** The Mw5.5 Pohang earthquake is shown by the red stars. The Pohang EGS is shown by the green squares. **a** Pohang EGS and its location in Korea (the inset map). The red circles represent 1-km distances from the site. **b, c** Hypocenters of the 2017 Mw5.5 earthquake and its aftershocks[65] (magenta circles with the event-magnitude scale (inset on **b**)). The aftershocks from the rectangle domain (black dashed contour on **b** and **d**) are shown projected on the vertical section **c** parallel to the dotted line in the centre of the rectangle domain. **c** Boreholes PX1 and PX2, the aftershocks and the Pohang Earthquake fault[65] (black dotted line). **d, e** Hypocenters of the Mw ≥ 0.9 fluid-injection-induced earthquakes (blue circles with the event-magnitude scale (inset on **b**)) at the Pohang EGS[18, 20]. **f** Timeline of fluid injection (black line), bleed-off[11, 18] (dashed green line), cumulative injected volume (magenta line) and the fluid-injection-induced earthquakes[18, 20] (blue circles) at the Pohang EGS.

termination of fluid injection (a finite time period after the termination of injection is characterised by a significant number of induced earthquakes[62,63]), because $L$ increases with $\Delta T$. This also contributes to the fact that $M_0^{\max} \propto \Delta T^{3/2}$ is more adequate for describing the 2017 Pohang Earthquake than $M_0^{\max} \propto \Delta V_f^{3/2}$.

$\Delta V_f$ is approximately proportional to the injection time. Therefore, $\Delta T$ and $\Delta V_f$ will be approximately interchangeable if the highest magnitude earthquake occurs before the termination of fluid injection (observed in a majority of the case histories). These quantities are indeed strongly correlated (Fig. 1). The scaling $L \propto \Delta T^{\xi}$ can then be approximately replaced by $L \propto \Delta V_f^{\xi}$, yielding $M_{\max} \approx 2\xi \log_{10} \Delta V_f + \text{const}$ and $M_0^{\max} \propto \Delta V_f^{3\xi}$. For example, we obtain a frequently reported observation[25,26], $M_0^{\max} \propto \Delta V_f^{3/2}$, for the case of normal-diffusion-controlled growth of the stimulated volume ($\xi = 1/2$, see please also Supplementary Information file 2, Eqs. 10 and 11).

In respect to the post-injection growth of the size $L$ of the stimulated volume, we should note that this volume will be relevant for earthquake triggering if corresponding perturbations of the pore-pressure and/or of the (poro)elastic stresses are still significant (e.g. above their fluctuations of tidal and seasonal nature). Thus, the scaling $M_{\max} - \Delta T$ has its natural limits. For example, in the case of a pressure-diffusion earthquake triggering, the seismogenic post-injection time period is of the order of the total duration time of the preceding injection operations, $t_0$, (see 'Methods' section). Then, the corresponding realistic scale of the spatial domain where the earthquake triggering will be probable is limited by the size of the order of several characteristic lengths $\sqrt{2t_0 D}$, where $D$ is a representative hydraulic diffusivity of hydraulic paths to the critically stressed faults.

**Monitoring the probability of a hypothetical $M_{\max}$.** Maximum induced magnitude and $\Delta T$ are unknown a priori parameters for

a particular fluid-injection experiment. However, one could estimate the exceedance probability of an assumed critical $M_{\max}$ during a given injection. The magnitudes of large runaway-induced ruptures are determined via the surrounding tectonic fault networks[23–25,64], with their frequency–magnitude distribution given by the Gutenberg–Richter statistic of the tectonic environment. However, their probability is enhanced due to the fluid injection. We take this into account by combining the occurrence probability of an earthquake with a magnitude $\geq M_{\max}$ with the seismogenic index model[23,30] (see 'Methods' section). This model contains the seismicity parameters, $\Sigma$ and $b$, which are functions of the seismotectonic features of the surrounding fault systems. A more or less seismically active fault system will potentially be activated if these parameters exhibit non-stationary behaviour during rock stimulation. Therefore, $\Sigma$ and $b$ should be monitored in real-time. We propose to monitor the worst-case probability based on the real-time evolution of these parameters:

$$W_{M \geq M_{\max}}(t) = 1 - \exp\left[-\Delta V_f(t) 10^{\sup \Sigma(t) - \inf b(t) M_{\max}}\right], \quad (1)$$

where we introduce two functions of the observation time $t$: $\sup \Sigma(t) = \max\{\Sigma(\tau) | 0 < \tau < t\}$ and $\inf b(t) = \min\{b(\tau) | 0 < \tau < t\}$. The combination of these two functions in the exponent on the right-hand side of Eq. (1) provides an estimate of the maximum occurrence probability of an earthquake with a magnitude $\geq M_{\max}$ at the current real-time $t$ counted from the start of injection operations (the time moment 0). Simultaneously, the quantity $W_{M \geq M_{\max}}(t)$ will be an upper-bound estimate of the occurrence probability after the timing of fluid-injection termination, if all injection operations are stopped at the time moment $t$ (see Methods). Thus, we call the quantity $W_{M \geq M_{\max}}(t)$ computed by Eq. (1) the worst-case exceedance probability. A statistically sound observation of the temporal behaviour of the $b$-value is challenging during the early stages of injections, especially when an imperfect monitoring system is used. A

potential alternative is to obtain a realistic estimate of the $b$-value from (a priori) available regional seismicity data. However, real-time monitoring of $\Sigma$ is feasible, such that the observation of significant events then becomes sufficient (see 'Methods' section and Supplementary Information file).

**2017 Pohang Earthquake and other case histories.** The epicentre of the 2017 Pohang Earthquake[18] was 510 m from the site of the Pohang-EGS project and ~15 km away to the north-northeast from the centre of Pohang, a city with a population of ~500,000. The earthquake occurred on 15 November 2017, after a series of water injections into two deep boreholes, PX-1 and PX-2, that were drilled into granodioritic rocks to 4215 and 4340 m depth, respectively (Fig. 2a–e). The hypocenters of the aftershocks of the Pohang Earthquake[65] are distributed approximately in the depth range between 2 and 7 km and in a lateral SSW-NNE zone of 11km long (Fig. 2b, c). The injection operations stimulated a one order of magnitude smaller rock volume (approximately of 1 km size) indicated by the induced seismicity[20] occurred before the Pohang Earthquake (Fig. 2d, e). Apparently, the hypocenter of the Pohang Earthquake was within the stimulated volume (corresponding to scenario ii of earthquake triggering described above).

We obtain the approximate lower and upper bounds of $\Sigma$ by treating the Pohang-EGS fluid injections in two different ways (see 'Methods' section). $\Sigma$ is approximately between $-2$ and $-1$ (Fig. 3). We observe the tendency for $\Sigma$ to increase with $t$. This tendency may be indicative of a gradual involvement of more seismically active domains in the stimulated volume. For example, there was

probably an expansion of the stimulated volume to a more seismogenic part of the major fault system that was initially intersected by borehole PX-2 and indicated by a massive mud loss during the drilling operations[18] in late 2015. Moreover, typical fault rocks were contained in the cuttings from borehole PX-2 next to the mud-loss depth interval[18,19]. Thus very likely, a fluid injection occurred nearly directly into a pre-existing large-scale fault.

The Mw3.3 event on 15 April 2017 was particularly alarming, as it indicated a possible sudden increase in $\Sigma$ to $-1$. Our estimates of $\Sigma$ are completely based on the seismicity data coming from the stimulated zone. Thus they are not representative for the total focal area of the Pohang Earthquake. Neither are they indicative for the final size of the Pohang Earthquake. The reasons of rupture stopping or rupture arrest can be illuminated by investigations of the rupture segmentation during the Pohang Earthquake sequence[66] (early aftershocks). It has been observed that the initial propagation of the main rupture segment and its subsidiary segment was likely arrested by two other fault segments, one to the northeast and another one to the southwest from the hypocenter domain. However, our estimates of $\Sigma$ indicate a dangerous tendency of non-stationarity in the temporal evolution of the seismogenic index, and thus an enhanced probability of triggering a runaway large-scale earthquake rupture. We use Eq. (1) to estimate such a probability of an assumed 5.5 magnitude earthquake (i.e. the probability of the Pohang Earthquake in this particular case history), which we denote below as $M_{\max}$. Please note that our estimates are not a prediction of a maximum possible earthquake magnitude but rather an estimate of the worst-case exceedance probability of an earthquake with the magnitude assumed. The probability

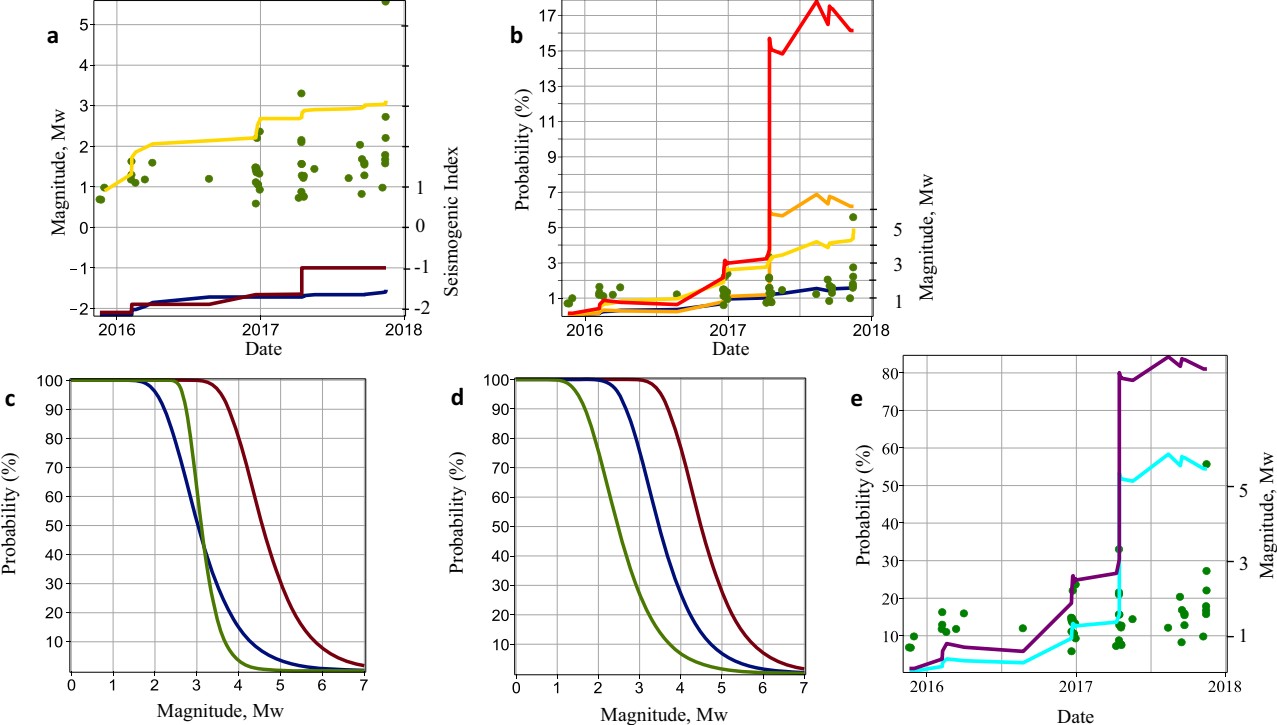

**Fig. 3 Magnitudes and $M_{\max}$ worst-case exceedance probabilities of induced earthquakes at Pohang. a, b, e** Mw $\geq$ 0.9 earthquakes (green points). **a** Expected[24] $M_{\max}$ (yellow line; see 'Methods' section), sup $\Sigma(t)$ assuming independent injection phases (dark-red line) and assuming the total stimulation period is a single continuous injection (dark-blue line). **b** Probabilities of a Mw5.5 earthquake computed from Eq. (1) with $b = 0.65$ (red and yellow lines) and $b = 0.73$ (orange and dark-blue lines) using both sup $\Sigma(t)$ estimates in **a**, respectively. **c, d** Probabilities of given magnitudes. **c** A Pohang-like situation: $\Delta V_f \approx 5850$ m$^3$, $b = 0.65$, $\Sigma = -1$ (red line) and $\Sigma = -2$ (blue line); and a Basel-like situation: $\Delta V_f \approx 11{,}570$ m$^3$, $b = 1.44$, $\Sigma = 0.25$ (green line). **d** Probabilities due to the termination of fluid injection based on magnitude thresholds: The Pohang-like situation in **c** (dark-red line), Mw3.3 (dark-blue line), and Mw2.3 (green line). **e** Probabilities of a stimulation-zone size event: the same as the red line in **b**, but for Mw4.0 (purple line) and Mw4.5 (cyan line).

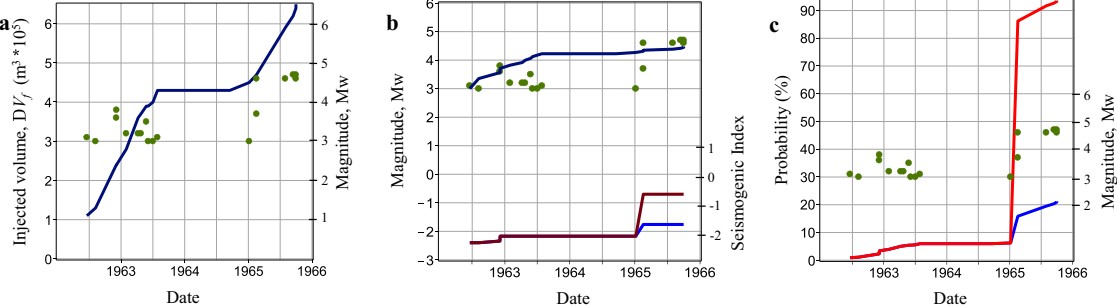

**Fig. 4 Mw ≥ 3 earthquakes (green points) at the Denver Rocky Mountain Arsenal wastewater injection site.** The earthquakes and injected volumes are taken from published data[4, 68]. **a** Earthquake occurrence versus cumulative injected fluid volume over time (dark-blue line). **b** Expected maximum magnitude[24] (dark-blue line), sup $\Sigma(t)$ assuming independent injection phases (dark-red line), and sup $\Sigma(t)$ assuming the total stimulation period is a single continuous injection (blue line). **c** Exceedance probability of Mw > 5.5 events at the Denver Rocky Mountain Arsenal wastewater injection site. The red and blue lines correspond to the upper and lower sup $\Sigma(t)$ estimates (**b**), respectively.

estimates of $M_{max} = 5.5$ (Fig. 3) correspond to the two known realistic values of the $b$-value[18,21], 0.65 and 0.73, which are also in a general agreement with the data on seismotectonic properties and zonation of the Korean Peninsula[67]. These probability estimates imply that all of the fluid injections in both boreholes should have been terminated after the Mw3.3 event. Furthermore, a better choice would have been to terminate the injection operations after the Mw2.3 event that was induced on 29 December 2016 by the fluid injections in PX-1. This would have kept the probability of a Mw5.5 event at approximately 3% or less (see Fig. 3b, the red line in the domain of the first quarter of 2017 and Fig. 3d, green line). The worst-case probability of the 2017 Pohang Earthquake was ultimately very significant, 15–17% (Fig. 3b–d, red and dark-red lines).

This worst-case probability was calculated using $\Sigma$, which characterises the stimulated zone, which is an order of magnitude smaller than the aftershock domain of the Pohang Earthquake. In the case of heterogeneous and/or unsteady tectonic stresses (the possibility of such a situation is indicated above by a non-stationarity of $\Sigma$; for example, the Mw5.4 Gyeongju earthquake[2,10,19] occurred on 12 September 2016 about 40 km south of the Pohang EGS could contribute to such a heterogeneous and/or unsteady stresses situation), the real probability of the Pohang Earthquake could be even higher. A strong indication of a possibility of a runaway rupture can also be seen from a very high worst-case probability of an event of the stimulation-zone size (this is a largest event for which Eq. (1) can be rigorously applied based on seismicity parameters observed in the stimulation zone). For the Pohang EGS, such an event has a magnitude of the order of Mw4–Mw4.5 (the rupture size of the order $10^3$m and the stress drop in the range 1–5 MPa). The temporal behaviour of the probability of a stimulation-zone size event during the Pohang-EGS-stimulation activity is similar to that calculated for the Pohang Earthquake (Fig. 3e). However, the probability of such an event has reached 50–80% (Fig. 3c, d dark-red line). Such a high probability of a stimulation-zone size event was already achieved immediately after the induced Mw3.3 event (Fig. 3e). However, there were no earthquakes of the strength between Mw3.3 and Mw5.5. This is an indication that a similar event may have become the 2017 Pohang Earthquake with an unstable runaway rupture. Thus, during the stimulation operations, the Mw2.3 event should be considered critical, and a safe stimulation strategy would be to keep the induced seismicity approximately below Mw2.0.

The low $b$-value and high $\Sigma$ increased the probability of the 2017 Pohang Earthquake. A comparison of the 2006 Basel EGS stimulation[12], which injected more fluid (~11,500 m³) than the

Pohang one, yielded a significantly higher corresponding $\Sigma$ (~0.25). However, the $b$-value (~1.44) was also higher, such that the probability of a Mw5.5 event at Basel was highly unlikely. However, the probability of a Mw3.4 event (such an earthquake led to the termination of the Basel EGS project[12]) was ~15% (Fig. 3c, green line). This level of probability could be predicted during the earlier stages of the Basel EGS project because an extremely high $\Sigma$ could be observed shortly after the onset of the stimulation (a very high $\Sigma$ remained nearly stationary throughout the injection period[31]).

The 1962–1968 Denver earthquakes[4,43,44,68] are another example of how the induced seismicity could be predicted, with the large $\Delta V_f$ being the main driver in this instance. Approximately 630,000 m³ of wastewater was injected down a single 3671-m-deep well that was drilled into the fractured Precambrian crystalline basement below the Rocky Mountain Arsenal[44]. The fluid injection began under pressure on 8 March 1962. It was interrupted from October 1963 to August 1964, and then resumed as a gravity flow in September 1964, with continued fluid injection under pressure commencing on 6 April 1965. Fluid injection was terminated on 20 February 1966 due to the induced seismicity[4,68] (Fig. 4a). We obtain an estimate of $\Sigma(t)$ for the Denver site by assuming a continuous injection experiment. This provides sup $\Sigma(t)$ in the −2.5 to −1.75 range (Fig. 4b). However, we obtain a sudden increase in the seismogenic index to −0.7 when we assume that the second injection period that began in September 1964 was an independent one (Fig. 4b). We compute the probability of a hypothetical Mw5.5 event (Fig. 4c). We use this magnitude for a direct comparison with the Pohang case study; corresponding to various sources[4,22,43,68] none of the Denver earthquakes were larger than Mw5.3 or even Mw4.85. We assume $b = 0.85$, based on the published data[4], and observe that even for the lower estimate of the seismogenic index, the probability of such a high magnitude event (>20%) was higher than that in Pohang (Fig. 4c). Furthermore, the probability was already high (~5%) before the beginning of the second phase of pressure-driven injections (Fig. 4c), and significant earthquakes (Mw4.6–4.7) had already begun to occur in early 1965. The largest earthquake[43] (≤Mw5.3) occurred on 9 August 1967.

The largest earthquakes in these three case histories occurred after the injection periods. The most critical parameters were a very high $\Sigma$ (with a moderate $\Delta V_f$ and high $b$-value) in Basel, a very low $b$-value (with a moderate to high $\Sigma$ and very moderate $\Delta V_f$) in Pohang, and an extremely high $\Delta V_f$ (with a moderate to high $\Sigma$ and nearly normal $b$-value) in Denver. The interplay between these parameters takes the form of Eq. (1). This equation forecasts a worst-case probability of an assumed maximum-

magnitude earthquake. It can be applied for a real-time monitoring of underground fluid-injection operations. This equation (along with our Eqs. (8) and (9)) can become a useful ingredient of traffic-light-type[3] or value-at-induced-risk-type[21] management approaches of injection-induced seismicity (for example, contributing to them by monitoring the worst-case probability of a critically large stimulation-zone size event as a proxy of the runaway rupture probability). The elapsed injection time $\Delta T$ is also an important implicit parameter of Eq. (1). The occurrence time of the Pohang Earthquake and its magnitude are in agreement with our proposed $\Delta T - M_{max}$ scaling for fluid-injection triggered seismicity (Fig. 1b, e). The nearly direct injection into the earthquake fault was likely a reason that a rather small injected fluid volume triggered a very large earthquake, in the upper right-hand domain of this scaling. Presumably and in agreement with this scaling, the Pohang Earthquake nucleation required a rather long time of pore-pressure-related perturbation propagation along the fault zone. This perturbation propagation was forced by multiple borehole fluid injections into the underground.

## Methods

**Seismogenic index and event probability**. We use the seismogenic index model[23,30,31] to analyse the case histories. $\Sigma$ quantifies the potential for an earthquake to occur at a particular geologic site due to fluid injection. This model has been used to characterise induced seismicity in various studies, such as water-disposal-related seismicity[8], seismicity triggered by hydraulic fracturing operations[16,69,70], and seismicity related to the Pohang-EGS activity[20].

The seismogenic index model[23,30] follows from the mass conservation of an approximately incompressible pore fluid (i.e. approximately pressure independent density). In the approximation of a monotonic non-decreasing injection pressure, this model describes the cumulative number of induced earthquakes, $N_{\geq M}(t)$, with moment magnitudes $\geq M$, that occur during the injection period $t$ as:

$$\log_{10} N_{\geq M}(t) = [\Sigma + \log_{10} \Delta V_f(t)] - bM. \quad (2)$$

The model assumes the Gutenberg–Richter statistic for the frequency of magnitude $\geq M$ induced events, $\log_{10} N_{\geq M} = a - bM$, where $a$ and $b$ are the parameters of the Gutenberg–Richter distribution. $\Sigma$ is the seismogenic index of the rock volume stimulated by the injection. Conventionally, $V_f(t)$ is given in $m^3$. $\Sigma$ can be estimated using Eq. (2) and a frequency–magnitude statistic of induced seismicity:

$$\Sigma = \log_{10} N_{\geq M}(t) - \log_{10} \Delta V_f(t) + bM. \quad (3)$$

We use this equation to compute $\Sigma$ in our analysis (Figs. 3 and 4b) (details of these computations are shown in the Supplementary Information file and the Supplementary Data 2 file). We calculate two estimates of $\Sigma$ in the Pohang case study to obtain approximate lower and upper bounds of this quantity as follows. We first neglect the non-monotonic character of the injection operations and estimate $\Sigma(t)$ directly by assuming that all of the injection phases are a single continuous injection experiment. This results in a somewhat underestimated $\Sigma$ (an approximate lower bound) because the fluid volume was also accounted for during the intervals where the pore pressure was lower than those reached in the previous injection phases. We then assume that all of the injection phases are independent experiments and compute the corresponding values of $\Sigma(t)$ (Supplementary Information file). This calculation results in a somewhat overestimated $\Sigma$ (an approximate upper bound) because some events that were induced by previous injection phases can influence the number of induced events in the later injection phases. During long-term bleed-off phases of the injected fluid, the relation between these two $\Sigma$ estimates can turn round because of possible late triggered events and resulting lower injected volumes at occurrence times of such events (Fig. 3a).

Equation (2) has been proposed[24] to compute an estimate of $M_{max}$. Substituting $N_{\geq M} = 1$ (this must be valid for the largest event in the observed earthquake series) into Eq. (2) yields[24] $M_{max} = \frac{1}{b}[\Sigma + \log_{10} \Delta V_f(t)]$. The $b$-value is usually close to 1, which means this relationship is very close to the result for the maximum arrested rupture[25]. Furthermore, this approach takes into account both the injected volume and seismogenic index that characterises the seismotectonic features of the injection site. However, the problem due to such an approach is inherited from the statistical nature of Eq. (2). If the observed events follow a Gutenberg–Richter magnitude distribution and the seismicity process can be approximated as a Poisson process (both are common realistic approximations), then the probability of the occurrence of a magnitude $M$ or larger event will be[30]:

$$W_{ev \geq M} = 1 - e^{-N_{\geq M}}. \quad (4)$$

We obtain $W_{ev \geq M_{max}} = 1 - e^{-1} \approx 0.632$ for the maximum expected event with $N_{\geq M} = 1$. This $M_{max}$ estimate, therefore, predicts a magnitude of an event, which

occurrence probability is 63% or even higher. It provides no constraints for larger magnitudes. If we use this estimate and express the term $[\Sigma + \log_{10} \Delta V_f(t)]$ from Eq. (2), we will obtain an equivalent most-probable type[24] of $M_{max}$ estimate: $M_{max} = M + \frac{1}{b}[\log_{10} N_{\geq M}(t)]$, where $M$ is the magnitude used in Eq. (3) for calculating $\Sigma$. This estimate of the expected $M_{max}$ is shown in Figs. 3a and 4b by the yellow and dark-blue lines, respectively.

The probability of an arbitrary magnitude event can be estimated by combining Eq. (4) with Eq. (2) to obtain[30]:

$$W_{ev \geq M}(t) = 1 - \exp[-V_f(t)10^{(\Sigma - bM)}]. \quad (5)$$

This equation provides estimates of the probability of triggered events with magnitudes $\geq M$ occurring in the time period from the beginning of fluid injection at the time moment 0 until the given time moment $t$.

**Event probability after injection termination**. A specific model of the seismicity triggering process is required to predict the seismicity rate after the termination of a fluid injection. One can calculate the seismicity rate after the termination of injection[63] by assuming both a triggering mechanism that is governed by pore-pressure diffusion and a constant injection rate into a homogeneous porous medium. The decay rate of the induced seismicity is similar to Omori's law, which describes the decay rate of aftershock activity after tectonic earthquakes, whereby the decay exponent, $p_d$, is larger than one[63]. The following approximation[63] yields the seismicity rate, $R_{ev}(t)$, after the termination of an injection at a constant rate (at least for $\tau_0 = O(1)$):

$$R_{ev}(t) \approx R_{ev}(t_0)\tau_0^{-p_d}, \quad (6)$$

where $t_0$ is the duration of the fluid injection and $\tau_0 = t/t_0$ denotes the normalised time after the termination of the injection ($\tau_0 \geq 1$).

The following approximation of $R_{ev}(t_0)$ for events with magnitudes $\geq M$ can be directly obtained from Eq. (2):

$$R_{ev}(t_0) \approx \frac{V_f(t_0)}{t_0}10^{\Sigma - Mb}. \quad (7)$$

Numerical computations[63] have shown that $p_d$ approaches 2 for long durations ($\tau_0$ on the order of 2 and larger). This is also supported by studies of the seismicity induced by large-scale massive fluid disposals in Oklahoma[8]. The estimated seismicity rates for the Fenton Hill and Soultz case studies[63] ($\tau_0$ on the order of 1) have provided high values of 7.5 and 9.5, respectively, for the $p_d$ exponent.

We obtain the following relationship when we use Eq. (6) for the events with magnitudes $\geq M$ that were triggered during the time range $[t_0, t]$ (fluid injection starts at the time moment 0):

$$\log_{10} N_{\geq M}(t_0, t) \approx [\Sigma + \log_{10} V_f(t_0)] - bM + \log_{10} \frac{1 - (t_0/t)^{p_d - 1}}{p_d - 1}. \quad (8)$$

Equation (8) simplifies to $\log_{10} N_{\geq M}(t_0, t) \approx [\Sigma + \log_{10} V_f(t_0)] - bM - \log_{10}(p_d - 1)$ in the limit of long time periods after the termination of fluid injection. We obtain $\log_{10} N_{\geq M}(t_0, t) \leq [\Sigma + \log_{10} V_f(t_0)] - bM$ for $p_d \geq 2$. Therefore, the event probability after the termination of fluid injection is roughly limited by its probability during the period $[t_0, t]$. The short elapsed time of $\Delta t$ after an injection termination causes the event number to increase as $\log_{10} N_{\geq M}(t_0, t) \approx [\Sigma + \log_{10} V_f(t_0)] - Mb + \log_{10}(\Delta t/t_0)$.

Let us assume that the injection was terminated immediately after a magnitude $M_{cr}$ event (assumed to be critical) that occurred at $t_0$. We can then use right-hand side of Eq. (3) with $N_{\geq M} = 1$ to compute the seismogenic index, which will provide the following estimate of the event number:

$$\log_{10} N_{\geq M}(M_{cr}, t) \approx b(M_{cr} - M) + \log_{10} \frac{1 - (t_0/t)^{p_d - 1}}{p_d - 1}, \quad (9)$$

when $\Sigma$ is either stationary or maximal at $t_0$ (the latter should usually be the case).

The event probabilities corresponding to Eqs. (8) and (9) are determined by substituting these equations into the left-hand side of Eq. (4), respectively (Fig. 3c, d).

## Data availability

The data supporting the findings of this study are provided in the manuscript, in the Supplementary Information and in the Supplementary Data 1 and Data 2 files. The data and the computation details underlying Fig. 1 are provided in the Supplementary Data 1 file and in the Supplementary Information file. The map outline in the inset of Fig. 2a was obtained using https://www.generic-mapping-tools.org/. The earthquake location data underlying Fig. 2 are available as supplementary information of Kim et al.[65] and Woo et al.[20]. The hydraulic and seismic data underlying Fig. 2f are available as the supplementary information and the source data of Yeo et al.[11]. The seismic and hydraulic data underlying Fig. 3 and corresponding computation details are provided in the Supplementary Information file and in the Supplementary Data 2 file. The corresponding hydraulic data are based on the source data of Yeo et al.[11]. The corresponding seismicity catalogue is based on the open-access Report of the Korean Government Commission[18]. The data underlying Fig. 4 are collected from plots and tables of the publications on the Denver case study[4,43,44,68]. These data and the computation details are provided in the Supplementary Information file.

## Code availability

We do not use any custom software. The details of our computations are given in the Supplementary Information file as prints of the corresponding Maple Worksheets.

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

## Acknowledgements

We thank the GSK and the sponsors of the PHASE consortium research project of Freie Universitaet Berlin for supporting the research presented in this paper. This work was also supported partly by the Nuclear Safety Research Program through the Korea Foundation of Nuclear Safety (KoFONS) using financial resources granted by the Nuclear Safety and Security Commission (NSSC) of the Republic of Korea (No. 1705010) and by the Korean Meteorological Administration Research Development Program (Grant No. KMI2018-02810).

## Author contributions

S.A.S. designed the research; S.A.S. and K.-H.K. conducted the research; S.A.S., K.-H.K. and J.-H.R. analysed the results and wrote the paper.

## Funding

## Competing interests

The authors declare no competing interests.
