## [Peer Review File · Nature Communications]

Magnitude and nucleation time of the 2017 Pohang Earthquake point to its predictable artificial triggeringReviewers' Comments:

Reviewer #1:

Remarks to the Author:

Please refer to attached review report.

Peer Review Report for:

Magnitude and nucleation time of the 2017 Pohang Earthquake point to its predictable artificial triggering (NCOMMS-21-12950-T)

S.A. Shapiro, K.H. Kim & J.H. Ree

Scope & Limitation of Review: My expertise is in the area engineered geologic fluid systems with specialization in nonisothermal multi-phase fluid flow, reservoir-scale hydromechanics and numerical simulation. Much of my research focuses on the hydraulics of injection-induced seismicity, but I am not a seismologist so there are some areas of this manuscript for which I am not qualified to comment.

General Comments: The subject manuscript presents an interesting application of the seismogenic index model for predicting maximum earthquake magnitude probability, e.g., forecasting worst-case earthquake scenarios at fluid injection sites. The method is built upon accepted, and well-documented methods, and it is shown to accurately predict the time-dependent magnitude exceedance probabilities for Pohang (ms. Fig. 3) and to a lesser extent for the RMA earthquake sequence (ms. Fig. 4). This is an important result that warrants prompt publication because it can be utilized for seismic monitoring at fluid injection sites worldwide.

From the perspective of injection hydraulics, I find the scaling dynamics between seismic moment (M_0) and injection time (ΔT) to be particularly interesting, but also a bit underdeveloped in the manuscript. Of course, the scaling relationship between M_0 and cumulative volume (ΔV) is well-documented from a phenomenological standpoint; however, the relationship between maximum magnitude (M_{\max}) and elapsed injection time is new. In my opinion, the time-dependent scaling relationship presented in this manuscript seems to more accurately reflect the underlying physics because pressure diffusion has predictable space-time scaling dynamics. I think the reader might benefit from a bit more discussion about the underlying physical processes that result in the ΔT - M_{\max} scaling dynamics, and why they are more predictive than ΔV - M_0 scaling dynamics for larger magnitude earthquakes.

Recommendation: Accept & publish promptly after minor revisions.

Minor Line Comments

Line 23: “normal-diffusion process” feels a bit technical for a general audience abstract because the alternative is anomalous diffusion, which, is very technical. Recommend just stating “pressure diffusion process.”

Lines 40 – 41: delete comma after “perturbation”; insert comma after “toughness”

Line 42: The proportionally constant is implied by all previous statements about proportional relationships, but introducing the symbol (γ) feels out of place because it’s only discussed qualitatively. Recommend putting γ into the proportional relationships or not using the symbol.

Line 89: Can a measure of spread be included in the analysis to quantify better ΔT - M_{\max} scaling.

Figures 1, 3 & 4: The minor grid lines are faint and difficult to use. Recommend deleting them. Also, axis labels are small and difficult to read, perhaps increase font?

Figure 2 has six panels, which are highly informative, but only panels a and f appear to be relevant to the figure reference at line 155-156. Perhaps panels b – e can be referenced in the passage between lines 152 – 170 to provide the reader additional context for the case study at Pohang?

Reviewer #2:

Remarks to the Author:

The hazard posed by induced and triggered seismicity is a growing concern for the geo-energy industry and the magnitude of the larger expected earthquake is a critical parameter for seismic hazard analysis. In the last 10-15 years a huge amount of work has been dedicated to improve our understanding of induced and/or triggered seismicity.

To forecast the maximum expected magnitude some studies have proposed that the seismic moment of the largest induced earthquake M_0^{\max} is proportional to the injected fluid volume, $M_0^{\max} \propto \Delta V_f$, e.g. McGarr, JGR, 2014.

Other studies, building on numerical simulations and fracture mechanics evaluated the moment of the largest arrested rupture that was induced by the pore-fluid pressure perturbation and proposed the relationship $M_0^{\max} \propto \Delta f^{(3/2)}$ (e.g. Galis et al., Science Advances, 2017).

The present manuscript analyses a comprehensive dataset of induced earthquakes and shows that the maximum magnitudes of these earthquakes scale with the logarithm of the elapsed time from the beginning of the fluid injection to the earthquake occurrence. From this observation the Authors: a) propose an alternative method to monitor and predict the exceedance probability of an assumed M_{\max} during fluid injection operation; b) show that this can be done in real time by monitoring the seismicogenic index (e.g., Shapiro et al., 2010).

The manuscript is well presented and scientifically sounds. The method proposed here will be extremely useful to predict the probability of larger earthquakes during fluid pressure stimulations and at the same time will push the geo-energy industry to develop high-resolution seismic networks to acquire real-time earthquake locations. The manuscript will be of broad impact for the scientific community since it represents a new and improved method to predict the maximum magnitude during fluid stimulations. Previous papers on these topics produced a large impact on the community as demonstrated by the large number of citations (McGarr, JGR, 2014 489 citations, Galis et al., Science Advances, 2017, 125 citations, Shapiro et al., 2010, 220 citations).

For all mentioned above, I consider the paper suitable for publication in Nature Geoscience.

Before publication I suggest to: a) better discuss one point; b) improve the description of the data presented in the figures; c) reinforce the last paragraph.

a) Paragraph M_{\max} scales with the log of the elapsed time.

At lines 118-120 the Authors say: "Large earthquakes have an enhanced occurrence probability at or shortly after the termination of fluid injection because L increases with ΔT ". This is well supported by the relationship between M_w and $\log \Delta T$ (for example Figure 1e). However, at some point with increasing time I suspect that this relationship does not hold anymore and therefore I think that a comment on this would be useful for the reader.

In addition, at the beginning of this paragraph there are some typos DT instead of ΔT .

b) I have found some difficulties in reading the text and collecting all the data presented in the figures. I suggest better describing the figures in the text. Here you are some examples.

Lines 82 and 83: The laboratory and mine-like observations (data points with $M_w < -3$, add in Fig. 1a and b). In the caption of figure 1, ...and observations obtained from mine (..) and laboratory experiments on small samples add $M_w < -3$).

The numbers of the x and y labels of the figure 1,2,4 are very small.

Figure 3.

The y label of 3b says: M_w & Probability (%). I do not understand why it is required to put M_w here. I suggest to use probability (%) in both 3b,c and d otherwise the reader is confused.

At lines 168-170 it is written: "This would have kept the probability of a $M_w 5.5$ event at approximately 3% or less. The worst-case probability of the 2017 Pohang Earthquake was ultimately very significant, 15%-17%". Better link the text with the figure, for example the last sentence should be related to figure 3b whereas the 3% prob should be related to 3d. How can the reader evaluate this 3% prob?

Line 176: Here the Authors say: "The probability of a Mw3.4 event (such an earthquake led to the termination of the Basel EGS project) was approximately 15% (Fig. 3). Add Fig. 3c green line. In Figure 4 the numbers of the ylabels are for both Mw and the cumulative fluid injected volume?"

c) In the last paragraph of the manuscript, lines 199-203, I have got the feeling that a sort of conclusions or wrap-up of the main finding of the manuscript is missing.

Sincerely
Cristiano Collettini

Reviewer #3:
Remarks to the Author:

I have reviewed the paper by S. A. Shapiro et al., entitled as "Magnitude and nucleation time of the 2017 Pohang Earthquake point to its predictable artificial triggering", submitted to Nature Communications (NCOMMS-21-12950-T). I have been extremely busy with testing three big rock deformation apparatuses, and I apologize for the delay in completing the review.

The 2017 Pohang earthquake was a triggered earthquake due to the EGS (enhanced geothermal system) activity that has received broad attention worldwide. I myself consider that this earthquake can be a prototype earthquake for detailed analysis of mechanisms of induced/triggered earthquakes because (i) natural seismicity was quite low in Pohang area before the EGS operations, (ii) seismicity and injection history of water are known well, (iii) induced earthquakes prior to the Pohang earthquake occurred in a narrow zone of about 1 km in size in granitic rocks (simple geology), (iv) the fault that moved during the Pohang earthquake was very close to the injection spot (within several hundred meters), and (v) fragment of fault rocks from the fault were recovered in borehole cuttings that allows measurements of frictional and transport properties of fault zone. The authors are a unique combination to search for the mechanism of the Pohang earthquake. The first author (SAS, a theoretical seismologist) has worked on many cases of induced earthquakes in the world and published many papers and two books on the subject. The second author (KHK) is one of the most active seismologists in Korea who recorded foreshocks, the main shock, and numerous aftershocks of the Pohang earthquakes using his own seismic network just above the EGS boreholes. The third author (JHR) is a leading structural geologist who has worked on tectonic and active faults in Korea. The third author proposed that the EGS activities caused the Pohang earthquake on the same day as the earthquake occurred. The authors applied the first author's method and estimated the maximum expected magnitude and its probability of the Pohang earthquake in the submitted paper. Their results are very important and such analyses can be done real time, and their paper will receive wide interest in diverse communities such as EGS, shale-gas production, CCS (carbon capture and sequestration), oil and gas productions, and reservoir-induced seismicity. I thus strongly recommend the acceptance of the paper after some revisions. The paper is written very clearly and logically with clear figures and a complete list of publications. However, unique features of the Pohang earthquake listed above (geological setting in particular) are not fully implemented in the paper and some assumptions may have to be re-evaluated as mentioned below.

My research areas are fault rocks and fault mechanics trying to understand earthquake mechanisms, and I have not done any seismological analyses such as those done in the submitted paper. As for the Pohang earthquake, I found lots of fragments of fault rocks that caused the Pohang earthquake in the borehole cuttings, as a member of the ORAC (Oversea Research Advisory Committee) of the Pohang Earthquake. Thus my comments and suggestions below are from geology and fault mechanics perspectives.

[1] Geological setting, EGS stimulated zone, and Pohang earthquake as a triggered event
As stated above, injection of water in Pohang was nearly direct injection into a large-scale pre-existing

fault that caused the Pohang earthquake. I think that this was a critical reason why injection of a rather small amount of water could cause the Pohang earthquake. Moreover, the size of stimulated zone as revealed by seismicity was ~ 1.0 km or slightly less and is consistent with the results of fluid-flow calculations (e.g., Figures 6 and 10 in Ellsworth et al., 2019, ref. 4 of submitted paper). Whereas the aftershocks are distributed little over 10 km (e.g., Figure 2b of the present paper). Distribution of aftershocks in the first three hours were about 5 km (Fig. 3 of Kim et al., 2018, ref. 2 in this paper). Focal area of the Pohang earthquake was distinctly larger than the EGS stimulated zone. Thus the ORAC Committee concluded that EGS activities induced earthquakes in the stimulated zone and that the induced earthquakes triggered the Pohang earthquake which propagated far beyond the stimulated zone. The epicenter of the Pohang earthquake was within the stimulated zone (the Pohang earthquake started within the stimulated zone), and this was critical for the conclusion. Seismicity outside of the stimulated zone was very, very low prior to the Pohang earthquake so that the Pohang earthquake was a natural earthquake, but it was triggered by the EGS activities. The Committee defined the earthquakes only in the stimulated zone as "induced earthquakes". I know that those features are consistent with what are mentioned in the submitted paper, but they should be emphasized and discussed in relation to the analysis of the present paper.

[2] Is the present analysis applicable to the whole focal area of the Pohang earthquake?

As I understand, the authors used the seismogenic index S and b parameter to quantify an increased level of seismicity due to water injection (S is in the a parameter of the Gutenberg-Richter equation). Data to determine S and b come from the stimulated zone, not from the outside zone, and to be precise those parameters are not usable to predict seismicity outside the stimulated zone. Thus I feel that the analysis is valid for estimating the probability of the largest size earthquake in the stimulated zone, not the earthquakes in the whole Pohang area. M3.2 event almost fully ruptured the stimulated zone, but it was still contained in the stimulated zone. The method will be useful to predict the minimum-size of an earthquake that can run away from the stimulated zone, but this is not a prediction of the M_{max} as attempted in the present paper. I hope that the authors clarify the problem.

[3] What stopped the Pohang earthquake rupture?

I have heard from seismologists that what stops earthquake rupture was a very difficult problem. I also know that scientists working on active faults often use geometrical bends, separation of faults, cross-cutting of faults, and zones of complex small fault system as segment boundaries (size of each segment determines M_{max} in the segment). Detailed paleoseismological work in trenches often supports such interpretations. The third author (JHR) has identified several subfaults in the aftershocks of the Pohang earthquake (I do not know if the result was published or not). Rupture of the Pohang earthquake could have propagated over a few subfaults, but the rupture could have stopped by hitting at other faults cutting the ruptured fault(s). Faults in southeast Korea are complex owing to overprinted faults during inversion tectonics as the third author showed long time ago, and this could be a reason why very large historical earthquakes did not occur in Korea. I think that the authors should add one or two paragraphs to address the problem.

[4] I found a few typo errors as list below.

Lines 75, 76m: "DT" to " ΔT " (Greek letter, two places), Also in Excel file: Table 2, row 1, column L

Line 328: "finland" to "Finland"

Line 349: "france" to "France"

I look forward to seeing the paper in print. I should be responsible with what I mention there and do not want to be anonymous.

Toshi with best regards,

Toshihiko Shimamoto

14th July, 2021

Dear Madams and Sirs,

This is our point-by point response to the reviewer's comments on our manuscript titled "**Magnitude and nucleation time of the 2017 Pohang Earthquake point to its predictable artificial triggering**" by Serge A. Shapiro, Kwang-Hee Kim and Jin-Han Ree. We greatly acknowledge the positive useful and constructive comments of all three reviewers. We agree with all these comments, and we have attempted to take them into account as complete as possible.

Our responses below and all changes in the manuscript are given in red colour.

General changes:

Our revised manuscript contains a detailed Data Availability section. During the revision, we found a most complete seismicity catalogue published by the Korean Government Commission on the Pohang Earthquake. We give a detailed reference to the corresponding web page in the Data Availability section. This catalogue includes 3 more induced events with Mw above 0.9 (there were 49 such events; such events are relevant for our analysis). We accounted for them in the revised manuscript. This produced only some nearly invisible changes in Figures 2f, 3a and 3b. All our conclusions, computations and results remained the same. They are completely stable and robust.

We have introduced some changes necessary to comply with the formatting instructions of the Journal. This refers particularly to the abstract and, to a small extent, to the introduction section. All our modifications are shown in the manuscript with colour highlighting.

Reviewer #1

Scope & Limitation of Review: My expertise is in the area engineered geologic fluid systems with specialization in nonisothermal multi-phase fluid flow, reservoir-scale hydromechanics and numerical simulation. Much of my research focuses on the hydraulics of injection-induced seismicity, but I am not a seismologist so there are some areas of this manuscript for which I am not qualified to comment. General Comments: The subject manuscript presents

an interesting application of the seismogenic index model for predicting maximum earthquake magnitude probability, e.g., forecasting worst-case earthquake scenarios at fluid injection sites. The method is built upon accepted, and well-documented methods, and it is shown to accurately predict the time-dependent magnitude exceedance probabilities for Pohang (ms. Fig. 3) and to a lesser extent for the RMA earthquake sequence (ms. Fig. 4). This is an important result that warrants prompt publication because it can be utilized for seismic monitoring at fluid injection sites worldwide. From the perspective of injection hydraulics, I find the scaling dynamics between seismic moment (M_0) and injection time (ΔT) to be particularly interesting, but also a bit underdeveloped in the manuscript. Of course, the scaling relationship between M_0 and cumulative volume (ΔV) is well-documented from a phenomenological standpoint; however, the relationship between maximum magnitude (M_{\max}) and elapsed injection time is new. In my opinion, the time-dependent scaling relationship presented in this manuscript seems to more accurately reflect the underlying physics because pressure diffusion has predictable space-time scaling dynamics. I think the reader might benefit from a bit more discussion about the underlying physical processes that result in the ΔT - M_{\max} scaling dynamics, and why they are more predictive than ΔV - M_0 scaling dynamics for larger magnitude earthquakes.

Recommendation: Accept & publish promptly after minor revisions

Response: Thank you indeed! We have inserted more discussion about the underlying physical processes that result in the ΔT - M_{\max} scaling dynamics on lines 115-116, 119-138 and 154-162 of the revised manuscript. On these lines, we have also attempted to additionally address why this scaling is more predictive than ΔV - M_0 one. Summarizing the explanations given on lines 119-162 of our manuscript one can tell that in contrast to the ΔV - M_0 scaling, the ΔT - M_{\max} scaling includes the stimulation dynamics after injection terminations. In addition, this scaling is a more direct indicator of the physics of the earthquake triggering processes. For example, the ΔV - M_0 scaling can be derived from the ΔT - M_{\max} scaling under several realistic restrictions (lines 147-153 of the revised manuscript).

Minor Line Comments

Line 23: “normal-diffusion process” feels a bit technical for a general audience abstract because the alternative is anomalous diffusion, which, is very technical. Recommend

just stating “pressure diffusion process.”

Response: We have revised the Abstract in agreement with the formatting requirements of the Journal. As consequence, we removed this from the Abstract and put more terminological explanations into the corresponding section of the text (lines 104, 105 of the revised manuscript).

Lines 40 – 41: delete comma after “perturbation”; insert comma after “toughness”

Response: To avoid the ambiguity we edited this sentence (lines 41, 42 of the revised manuscript).

Line 42: The proportionally constant is implied by all previous statements about proportional relationships, but introducing the symbol (γ) feels out of place because it’s only discussed qualitatively. Recommend putting γ into the proportional relationships or not using the symbol.

Response: We removed the symbol and edited the lines 42 and 48 of the revised manuscript.

Line 89: Can a measure of spread be included in the analysis to quantify better ΔT - M_{max} scaling.

Response: Yes. Lines 92-95 of the revised manuscript read: “For example, a linear regression of the ΔT - M_{max} scaling provides significantly better statistics than the equivalent fitting of the ΔV_f – M_{max} scaling (e.g., a significantly higher coefficient of determination and significantly lower standard deviations of fitting parameters (Supplementary File 3, equations 10 and 11)).”

Figures 1, 3 & 4: The minor grid lines are faint and difficult to use. Recommend deleting them. Also, axis labels are small and difficult to read, perhaps increase font?

Response: Done.

Figure 2 has six panels, which are highly informative, but only panels a and f appear to be relevant to the figure reference at line 155-156. Perhaps panels b – e can be referenced in the passage between lines 152 – 170 to provide the reader additional context for the case study at Pohang.

Response: This has been improved on lines 134, 189, 191 and 193 of the revised manuscript.

Reviewer #2

The hazard posed by induced and triggered seismicity is a growing concern for the geo-energy industry and the magnitude of the larger expected earthquake is a critical parameter for seismic hazard analysis. In the last 10-15 years a huge amount of work has been dedicated to improve our understanding of induced and/or triggered seismicity.

To forecast the maximum expected magnitude some studies have proposed that the seismic moment of the largest induced earthquake M_0^{\max} is proportional to the injected fluid volume, $M_0^{\max} \propto \Delta V_f$, e.g. McGarr, JGR, 2014.

Other studies, building on numerical simulations and fracture mechanics evaluated the moment of the largest arrested rupture that was induced by the pore-fluid pressure perturbation and proposed the relationship $M_0^{\max} \propto \Delta_f^{(3/2)}$ (e.g. Galis et al., Science Advances, 2017).

The present manuscript analyses a comprehensive dataset of induced earthquakes and shows that the maximum magnitudes of these earthquakes scale with the logarithm of the elapsed time from the beginning of the fluid injection to the earthquake occurrence. From this observation the Authors: a) propose an alternative method to monitor and predict the exceedance probability of an assumed M_{\max} during fluid injection operation; b) show that this can be done in real time by monitoring the seismogenic index (e.g., Shapiro et al., 2010).

The manuscript is well presented and scientifically sounds. The method proposed here will be extremely useful to predict the probability of larger earthquakes during fluid pressure stimulations and at the same time will push the geo-energy industry to develop high-resolution seismic networks to acquire real-time earthquake locations. The manuscript will be of broad impact for the scientific community since it represents a new and improved method to predict the maximum magnitude during fluid stimulations. Previous papers on these topics produced a large impact on the community as demonstrated by the large number of citations (McGarr, JGR, 2014 489 citations, Galis et al., Science Advances, 2017, 125 citations, Shapiro et al., 2010, 220 citations).

For all mentioned above, I consider the paper suitable for publication in Nature Geoscience.

Before publication I suggest to: a) better discuss one point; b) improve the description of the data presented in the figures; c) reinforce the last paragraph.

Response: Thank you very much for a very positive and useful review! Below we

are giving our responses to your comments.

a) Paragraph Mmax scales with the log of the elapsed time.

At lines 118-120 the Authors say: "Large earthquakes have an enhanced occurrence probability at or shortly after the termination of fluid injection because L increases with ΔT ". This is well supported by the relationship between M_w and $\log \Delta T$ (for example Figure 1e). However, at some point with increasing time I suspect that this relationship does not hold anymore and therefore I think that a comment on this would be useful for the reader.

In addition, at the beginning of this paragraph there are some typos DT instead of ΔT .

Response: We comment on this point on lines 155-162 of the revised manuscript as follows: "In respect to the post-injection growth of the size L of the stimulated volume, we should note that this volume will be relevant for earthquake triggering if corresponding perturbations of the pore pressure and/or of the (poro)elastic stresses are still significant (e.g., above their fluctuations of tidal and seasonal nature). Thus, the scaling $M_{max}-\Delta T$ has its natural limits. For example, in the case of a pressure-diffusion earthquake triggering, the seismogenic post-injection time period is of the order of the total duration time of the preceding injection operations, t_0 , (see the Methods). Then, the corresponding realistic scale of the spatial domain where the earthquake triggering will be probable is limited by the size of the order of several characteristic lengths $\sqrt{2t_0D}$, where D is a representative hydraulic diffusivity of hydraulic paths to the critically stressed faults."

b) I have found some difficulties in reading the text and collecting all the data presented in the figures. I suggest better describing the figures in the text. Here you are some examples.

Lines 82 and 83: The laboratory and mine-like observations (data points with $M_w < -3$, add in Fig. 1a and b). In the caption of figure 1,and observations obtained from mine (..) and laboratory experiments on small samples add $M_w < -3$).

Response: We have introduced all these changes and attempted to better describe the figures in the text: please see lines 84, 85, 134, 189, 191, 193, 267, 268, 290, 300-302, 316-320, and the captions of Fig 1,3.

The numbers of the x and y labels of the figure 1,2,4 are very small.

Figure 3.

The y label of 3b says: M_w & Probability (%). I do not understand why it is required to put M_w here.

I suggest to use probability (%) in both 3b, c and d otherwise the reader is confused.

Response: All suggested changes have been introduced to the revised Figures 1, 3 and 4. The labels of the axes have been decoupled.

At lines 168-170 it is written: "This would have kept the probability of a Mw5.5 event at approximately 3% or less. The worst-case probability of the 2017 Pohang Earthquake was ultimately very significant, 15%–17%". Better link the text with the figure, for example the last sentence should be related to figure 3b whereas the 3% prob should be related to 3d. How can the reader evaluate this 3% prob?

Line 176: Here the Authors say: "The probability of a Mw3.4 event (such an earthquake led to the termination of the Basel EGS project) was approximately 15% (Fig. 3). Add Fig. 3c green line.

In Figure 4 the numbers of the y labels are for both Mw and the cumulative fluid injected volume?

Response: All suggested changes have been introduced to the revised Figures 3 and 4. The labels of the axes have been decoupled. In the revised manuscript, the corresponding referencing to the panels of Fig 3 are given on lines 226-228 as follows: "This would have kept the probability of a Mw5.5 event at approximately 3% or less (see Fig. 3b, the red line in the domain of the first quarter of 2017 and Fig. 3d, green line). The worst-case probability of the 2017 Pohang Earthquake was ultimately very significant, 15%–17% (Fig. 3 b, c, d, red and dark-red lines)." The requested change in the reference to Fig 3c also was introduced on line 234.

c) In the last paragraph of the manuscript, lines 199-203, I have got the feeling that a sort of conclusions or wrap-up of the main finding of the manuscript is missing.

Response: We have expanded this paragraph. These are lines 257-272 of the revised manuscript. We hope that you will find it now acceptable.

Reviewer #3

Dear Dr. Mueller

May 12, 2021

I have reviewed the paper by S. A. Shapiro et al., entitled as "Magnitude and nucleation time of the 2017 Pohang Earthquake point to its predictable artificial triggering", submitted to Nature Communications (NCOMMS-21-12950-T). I have been extremely busy with testing three big rock deformation apparatuses, and I apologize for the delay in completing the review.

The 2017 Pohang earthquake was a triggered earthquake due to the EGS (enhanced geothermal system) activity that has received broad attention worldwide. I myself consider that this earthquake can be a prototype earthquake for detailed analysis of mechanisms of induced/triggered earthquakes because (i) natural seismicity was quite low in Pohang area before the EGS operations, (ii) seismicity and injection history of water are known well, (iii) induced earthquakes prior to the Pohang earthquake occurred in a narrow zone of about 1 km in size in granitic rocks (simple geology), (iv) the fault that moved during the Pohang earthquake was very close to the injection spot (within several hundred meters), and (v) fragment of fault rocks from the fault were recovered in borehole cuttings that allows measurements of frictional and transport properties of fault zone. The authors are a unique combination to search for the mechanism of the Pohang earthquake. The first author (SAS, a theoretical seismologist) has worked on many cases of induced earthquakes in the world and published many papers and two books on the subject. The second author (KHK) is one of the most active seismologists in Korea who recorded foreshocks, the main shock, and numerous aftershocks of the Pohang earthquakes using his own seismic network just above the EGS boreholes. The third author (JHR) is a leading structural geologist who has worked on tectonic and active faults in Korea. The third author proposed that the EGS activities caused the Pohang earthquake on the same day as the earthquake occurred. The authors applied the first author's method and estimated the maximum expected magnitude and its probability of the Pohang earthquake in the submitted paper. Their results are very important and such analyses can be done real time, and their paper will receive wide interest in diverse communities such as EGS, shale-gas production, CCS (carbon capture and sequestration), oil and gas productions, and reservoir-induced seismicity. I thus strongly recommend the acceptance of the paper after some revisions. The paper is written very clearly and logically with clear figures and a complete list of publications. However, unique features of the Pohang earthquake listed above (geological setting in particular) are not fully implemented in the paper and some assumptions may have to be re-evaluated as mentioned below.

My research areas are fault rocks and fault mechanics trying to understand earthquake mechanisms, and I have not done any seismological analyses such as those done in the submitted paper. As for the Pohang earthquake, I found lots of fragments of fault rocks that caused the Pohang earthquake in the borehole cuttings, as a member of the ORAC (Oversea Research Advisory Committee) of the Pohang Earthquake. Thus my comments and suggestions below are from geology and fault mechanics perspectives.

Response: Thank you very much! Below we are giving our responses to your comments.

[1] Geological setting, EGS stimulated zone, and Pohang earthquake as a triggered event

As stated above, injection of water in Pohang was nearly direct injection into a large-scale pre-existing fault that caused the Pohang earthquake. I think that this was a critical reason why injection of a rather small amount of water could cause the Pohang earthquake. Moreover, the size of stimulated zone as revealed by seismicity was ~ 1.0 km or slightly less and is consistent with the results of fluid-flow calculations (e.g., Figures 6 and 10 in Ellsworth et al., 2019, ref. 4 of submitted paper). Whereas the aftershocks are distributed little over 10 km (e.g., Figure 2b of the present paper). Distribution of aftershocks in the first three hours were about 5 km (Fig. 3 of Kim et al., 2018, ref. 2 in this paper). Focal area of the Pohang earthquake was distinctly larger than the EGS stimulated zone. Thus the ORAC Committee concluded that EGS activities induced earthquakes in the stimulated zone and that the induced earthquakes triggered the Pohang earthquake which

propagated far beyond the stimulated zone. The epicenter of the Pohang earthquake was within the stimulated zone (the Pohang earthquake started within the stimulated zone), and this was critical for the conclusion. Seismicity outside of the stimulated zone was very, very low prior to the Pohang earthquake so that the Pohang earthquake was a natural earthquake, but it was triggered by the EGS activities. The Committee defined the earthquakes only in the stimulated zone as "induced earthquakes". I know that those features are consistent with what are mentioned in the submitted paper, but they should be emphasized and discussed in relation to the analysis of the present paper.

Response: We agree with all these points and attempted to account for this comment on the following lines of the revised manuscript: 123-136 as follows:

"The following three scenarios²³ can be relevant for triggering Mmax earthquakes: (i) the rupture surface of the earthquake is contained within the stimulated volume (sometimes

strictly such events are called induced earthquakes^{18, 19, 23}); or (ii) the nucleation point of the rupture surface and its significant part are contained within the stimulated volume; or finally, (iii) the rupture surface is just touched or somewhat intersected by the stimulated volume (sometimes events, addressed in the scenarios (ii) and (iii) are called triggered ones^{18, 19, 23}). The cases of ruptures arrested inside or outside of the stimulated volume as well as runaway ruptures are parts of these scenarios. At least in the scenarios (ii) and (iii), magnitudes of triggered earthquakes are very likely controlled by tectonic features of corresponding geological sites rather than by the scale L of the stimulated volume. However, the occurrence probability of such earthquakes depends on this scale, as it follows from our consideration above. The Pohang Earthquake is an example of such an event. Figure 2 a-e indicates that it corresponds to the scenario ii. This scenario is also in agreement with the conclusion of the ORAC Committee¹⁸ that the Pohang Earthquake was triggered by an EGS-stimulation impact on the earthquake hypocenter domain. This impact could have various forms, for example a direct injection-produced pore pressure perturbation or a combined pore pressure and stress perturbations¹¹ of induced earthquakes in the stimulated volume.”

... and further on lines 189-195 of the revised manuscript:

“The hypocenters of the aftershocks of the Pohang Earthquake⁶⁵ are distributed approximately in the depth range between 2 and 7 km and in a lateral SSW-NNE zone of 11km long (Fig. 2 b,c). The injection operations stimulated one order of magnitude smaller rock volume (approximately of 1km size) indicated by the induced seismicity²⁰ occurred before the Pohang Earthquake (Fig. 2 d, e). Apparently, the hypocenter of the Pohang Earthquake was within the stimulated volume (corresponding to scenario ii of earthquake triggering described above).”

...and further on lines 198-204 of the revised manuscript:

“We observe the tendency for Σ to increase with t . This tendency may be indicative of a gradual involvement of more seismically active domains in the stimulated volume. For example, there was probably an expansion of the stimulated volume to a more seismogenic part of the major fault system that was initially intersected by borehole PX-2 and indicated by a massive mud loss during the drilling operations¹⁸ in late 2015. Moreover, typical fault rocks were contained in the cuttings from borehole PX-2 next to the mud-loss depth interval^{18, 19}. Thus very likely, a fluid injection occurred nearly directly into a pre-existing large-scale fault.”

...and finally on lines 267-272 of the revised manuscript:

“The occurrence time of the Pohang Earthquake and its magnitude are in agreement with our proposed ΔT - M_{\max} scaling for fluid-injection triggered seismicity (Figure 1b,e). The nearly direct injection into the earthquake fault was likely a reason that a rather small injected fluid volume triggered a very large earthquake, in the upper right-hand part of this scaling. Presumably and in agreement with this scaling, the Pohang Earthquake nucleation required a rather long time of pore-pressure-related perturbation propagation along the fault zone. This perturbation propagation was forced by multiple borehole fluid injections into the underground.”

[2] Is the present analysis applicable to the whole focal area of the Pohang earthquake? As I understand, the authors used the seismogenic index S and b parameter to quantify an increased level of seismicity due to water injection (S is in the a parameter of the Gutenberg-Richter equation). Data to determine S and b come from the stimulated zone, not from the outside zone, and to be precise those parameters are not usable to predict seismicity outside the stimulated zone. Thus I feel that the analysis is valid for estimating the probability of the largest size earthquake in the stimulated zone, not the earthquakes in the whole Pohang area. M3.2 event almost fully ruptured the stimulated zone, but it was still contained in the stimulated zone. The method will be useful to predict the minimum-size of an earthquake that can run away from the stimulated zone, but this is not a prediction of the M_{\max} as attempted in the present paper. I hope that the authors clarify the problem.

Response: We hope that we contributed to the clarification of this point on the lines 206-209 of the revised manuscript by the following:

“Our estimates of Σ are completely based on the seismicity data coming from the stimulated zone. Thus, they are not representative for the total focal area of the Pohang Earthquake. Neither are they indicative for the final size of the Pohang Earthquake...”

... and further on lines 213-219 of the revised manuscript:

“However, our estimates of Σ indicate a dangerous tendency of non-stationarity in the temporal evolution of the seismogenic index, and thus an enhanced probability of triggering a runaway large-scale earthquake rupture. We use equation 1 to estimate such a probability of an assumed 5.5 magnitude earthquake (i.e., the probability of the Pohang Earthquake in this particular case history), which we denote below as M_{\max} . Please note that our estimates are not a prediction of a maximum possible earthquake magnitude but rather an estimate of the worst-case exceedance probability of an earthquake with the magnitude assumed.”

...and finally on lines 261-265 of the revised manuscript by the following:

“This equation forecasts a worst-case probability of an assumed maximum-magnitude earthquake. It can be applied for a real-time monitoring of underground fluid-injection operations. This equation (along with our equations 8 and 9) can become a useful ingredient of traffic-light-type³ or value-at-induced-risk-type²¹ management approaches of injection induced seismicity.”

[3] What stopped the Pohang earthquake rupture?

I have heard from seismologists that what stops earthquake rupture was a very difficult problem. I also know that scientists working on active faults often use geometrical bends, separation of faults, cross-cutting of faults, and zones of complex small fault system as segment boundaries (size of each segment determines M_{max} in the segment). Detailed paleoseismological work in trenches often supports such interpretations. The third author (JHR) has identified several subfaults in the aftershocks of the Pohang earthquake (I do not know if the result was published or not). Rupture of the Pohang earthquake could have propagated over a few subfaults, but the rupture could have stopped by hitting at other faults cutting the ruptured fault(s). Faults in southeast Korea are complex owing to overprinted faults during inversion tectonics as the third author showed long time ago, and this could be a reason why very large historical earthquakes did not occur in Korea. I think that the authors should add one or two paragraphs to address the problem.

Response: This is indeed a very important and unanswered seismologic question. Accounting for the fact that in spite of its importance it is somewhat outside of the main task of our research, which cannot contribute to this aspect significantly, we added the following comment on lines 208-213 of the revised manuscript:

“The reasons of rupture stopping or rupture arrest can be illuminated by investigations of the rupture segmentation during the Pohang Earthquake sequence⁶⁶ (early aftershocks). It has been observed that the initial propagation of the main rupture segment and its subsidiary segment was likely arrested by two other fault segments, one to the northeast and another one to the southwest from the hypocenter domain.”

[4] I found a few typo errors as list below.

Lines 75, 76m: "DT" to " ΔT " (Greek letter, two places), Also in Excel file: Table 2, row 1, column L

Line 328: "finland" to "Finland"

Line 349: "france" to "France"

Response: Done. Thank you indeed!

Sincerely yours,

Serge A. Shapiro, Kwang-Hee Kim and Jin-Han Ree

Corresponding author:

Serge A. Shapiro,
Earth Science Department, Freie Universität Berlin, 12249, Malteserstr. 74-100, Building D,
12249, Berlin, Germany, E-mail: shapiro@geophysik.fu-berlin.de, Mobile: + 49-1716495415

Reviewers' Comments:

Reviewer #2:

Remarks to the Author:

I have already considered the manuscript a great scientific contribution in my first interaction with it. During the revision the Authors significantly improved some details that in my view are useful to better clarify some points. Therefore, I strongly support the publication of the manuscript in Nature Communications.

In reading the manuscript I have found some typos and therefore I suggest to carefully check text/figures of the manuscripts. Here you are the one I have found at lines 90-91. "However, magnitude scaling with ΔT describes M_{max} better by providing a more compact set of data points (Fig. 1c,d)".

I think that should be (Fig. 1c,E).

Really nice work! I have enjoyed the reading and learnt a lot from it.

Sincerely

Cristiano Collettini

Reviewer #3:

Remarks to the Author:

Dr. Sophis Rasheed, Staff, Nature Communications

Dear Editor of NCOMMS

August 14, 2021

I have re-reviewed the revised paper by S. A. Shapiro et al., entitled as "Magnitude and nucleation time of the 2017 Pohang Earthquake point to its predictable artificial triggering", submitted to Nature Communications (NCOMMS-21-12950-T). The authors responded nearly all comments and suggestions of reviewers and revised the manuscript very nicely. The paper may be acceptable as it is. However, the paper will have a big impacts to several research areas related to deep drilling, and I think that the paper can be improved further to give an insight on safe drilling.

[1] Applicability of the seismicity data during the Pohang EGS activity

I pointed out in my review that the Pohang earthquake was a triggered event whose rupture propagated far beyond the stimulate zone (~ 1 km) and that seismicity parameters such as Σ and b from the Pohang earthquake sequence are not applicable to earthquakes outside of the stimulated zone. This limitation is clearly mentioned in the revised manuscript (Lines 205-213). However, essentially the same estimate of probability of the occurrence of the Pohang earthquake is presented as a core of the paper as in the first draft, under a term of "the worst-case exceedance probability of an earthquake with the magnitude assumed" (Lines 218-219). Of course, I understood that $M_{max} = 5.5$ (magnitude of the Pohang earthquake) was assumed, not predicted from the analysis. What I worry is the following. The probability will be higher if one assumes a smaller magnitude that correspond to the size of stimulated zone (~ 1 km) than the reported probability for the Pohang earthquake in the paper. If the earthquake rupture runs away from the stimulated zone, we have no constraint on when it stops because a triggered event is a natural earthquake. This aspect of natural earthquakes is clearly stated on Lines 208-213 in the revised earthquake. Gyeongju earthquake of Mw 5.4, a natural earthquake, occurred about 40 km south of Pohang about a year prior to the Pohang earthquake, and tectonic stress was likely to have been high enough to allow the rupture propagation of the Pohang earthquake. The point I want to make is that the real probability for the Pohang

earthquake could have been that for a stimulation-zone size earthquake, rather than that for the Pohang-size earthquake. Moreover, a triggered event could have been larger than the Pohang earthquake if stress and geological setting allowed although the authors will get a lower probability if a larger M_{max} is assumed.

It is fine to keep all results in the paper, but (1) the term, "the worst-case exceedance probability" has to be defined or explained, and (2) such an ambiguity in estimating the probability of earthquakes beyond the stimulated zone has to be explained in the paper.

[2] Implications for safe drilling and fluid injection

Operation of deep drilling with fluid injection without inducing/triggering damaging earthquake is very important now and will be even more important in the future. The analysis proposed in this paper can be done real time with the drilling operation and may lead to a renovation in the safe drilling, as mentioned clearly on Lines 261-265. The authors calculated a probability of the Pohang-size earthquake in the present paper ($M_{max} = 5.5$). However, I strongly feel that only this demonstration is weak as a recommendation for safe drilling operation. The stimulation-zone size earthquake is the limit of M_{max} for which its probability can be estimated rigorously from the seismicity parameters for the Pohang earthquake sequence. Also, the probability for this M_{max} can be taken as a probability for a triggered earthquake that can propagate beyond the stimulated zone. Such an event may turn out to be small, but it may grow to the size of the Pohang earthquake or even larger. Thus, avoiding earthquakes beyond the stimulated zone can be a useful guide for safe drilling operation at least for the size of the stimulated zone in Pohang (M_{max} has to be set up to an acceptable limit when the stimulated zone becomes very large). It will be very, very useful for future drilling to construct a new diagram similar to Figure 3 using M_{max} corresponding to the size of the stimulation zone, demonstrating what could have been safe drilling operation in Pohang avoiding triggered events. A point of interest is whether or not any drilling operation possible without inducing the stimulation-zone size earthquake after M3.3 event. Result on Denver earthquake in Figure 4 are interesting and important, but it can be reported as a new and comprehensive paper elsewhere (information given in this paper is not enough!).

[3] "Probability" of earthquake in Figure 3 and in the main text

A small comment on a term "probability". In Japan, a period is always specified in a probability of an earthquake (e.g., probability in the next 30 years). So I wondered what period is referred to the probability in Figure 3 of the present paper. Lines 324-325 in Methods gives "probability of triggered events with magnitudes $\geq M$ occurring in the time period from the beginning of fluid injection until injection time t ". "Injection time" is not specified (there were multiple injections) and it should be "time" on the horizontal axis of Figure 3. I thus consider the probability as an instantaneous probability at a give time. Please clarify this and adding a definition on "probability" in figure caption will be useful to readers.

I understand that reviewers should not raise new issues that were not commented in the first review in re-review of a paper. The above comments [1] and [2] may appear to be new, but it came from my initial comment that the authors' method is applicable only to earthquakes up to the size of the stimulated zone. I hope that the authors take those as constructive to make a good paper even better. I suggest publication of the paper after some revisions. I look forward to seeing the paper in print whatever the final version is.

Toshi with best regards,

Toshihiko Shimamoto

2nd September, 2021

Dear Madams and Sirs,

This is our point-by-point response to the reviewer's comments on the 1st revised version our manuscript titled "**Magnitude and nucleation time of the 2017 Pohang Earthquake point to its predictable artificial triggering**" by Serge A. Shapiro, Kwang-Hee Kim and Jin-Han Ree. We greatly acknowledge the positive, useful and constructive comments of both reviewers #2 and #3. We agree with nearly all these comments, and we have attempted to take them as complete as possible into account.

Below, we give our responses (as well as the corresponding changes in the manuscript) in red colour.

Reviewer #2

I have already considered the manuscript a great scientific contribution in my first interaction with it. During the revision the Authors significantly improved some details that in my view are useful to better clarify some points. Therefore, I strongly support the publication of the manuscript in Nature Communications.

In reading the manuscript I have found some typos and therefore I suggest to carefully check text/figures of the manuscripts. Here you are the one I have found at lines 90-91. "However, magnitude scaling with ΔT describes Mmax better by providing a more compact set of data points (Fig. 1c,d)".

I think that should be (Fig. 1c,E).

Really nice work! I have enjoyed the reading and learnt a lot from it.

Response: Thank you indeed! We have corrected this and other found typos and attempted to improve some formulations (lines 91, 314, 315, 353, 354, 381, 391).

Reviewer #3

I have re-reviewed the revised paper by S. A. Shapiro et al., entitled as "Magnitude and nucleation time of the 2017 Pohang Earthquake point to its predictable artificial triggering", submitted to Nature Communications (NCOMMS-21-12950-T). The authors responded nearly all comments and suggestions of reviewers and revised the

manuscript very nicely. The paper may be acceptable as it is. However, the paper will have a big impacts to several research areas related to deep drilling, and I think that the paper can be improved further to give an insight on safe drilling.

Response: Thank you indeed! Your comments and suggestions are very useful and we have attempted to incorporate nearly all of them into the revised version. See please the details below.

[1] Applicability of the seismicity data during the Pohang EGS activity

I pointed out in my review that the Pohang earthquake was a triggered event whose rupture propagated far beyond the stimulate zone (~ 1 km) and that seismicity parameters such as Σ and b from the Pohang earthquake sequence are not applicable to earthquakes outside of the stimulated zone. This limitation is clearly mentioned in the revised manuscript (Lines 205-213). However, essentially the same estimate of probability of the occurrence of the Pohang earthquake is presented as a core of the paper as in the first draft, under a term of "the worst-case exceedance probability of an earthquake with the magnitude assumed" (Lines 218-219). Of course, I understood that $M_{\max} = 5.5$ (magnitude of the Pohang earthquake) was assumed, not predicted from the analysis. What I worry is the following. The probability will be higher if one assumes a smaller magnitude that correspond to the size of stimulated zone (~ 1 km) than the reported probability for the Pohang earthquake in the

paper. If the earthquake rupture runs away from the stimulated zone, we have no constraint on when it stops because a triggered event is a natural earthquake. This aspect of natural earthquakes is clearly stated on Lines 208-213 in the revised earthquake. Gyeongju earthquake of Mw 5.4, a natural earthquake, occurred about 40 km south of Pohang about a year prior to the Pohang earthquake, and tectonic stress was likely to have been high enough to allow the rupture propagation of the Pohang earthquake. The point I want to make is that the real probability for the Pohang earthquake could have been that for a stimulation-zone size earthquake, rather than that for the Pohang-size earthquake. Moreover, a triggered event could have been larger than the Pohang earthquake if stress and geological setting allowed although the authors will get a lower probability if a larger M_{\max} is assumed.

It is fine to keep all results in the paper, but (1) the term, "the worst-case exceedance probability" has to be defined or explained, and (2) such an ambiguity in estimating the probability of earthquakes beyond the stimulated zone has to be explained in the paper.

Response: This comment and the next one are very closely related. Correspondingly, to some extent, the changes we introduced address them both.

The definition of the worst-case exceedance probability has been given on lines 178-184 of the second revision of the manuscript. As for the ambiguity and applicability of this estimate for the Pohang Earthquake and corresponding implications for safe EGS fluid stimulations (your next comment), we address these points on lines 234 – 253 of the second revision as follows:

“This worst-case probability was calculated using Σ , which characterizes the stimulated zone, which is an order of magnitude smaller than the aftershock domain of the Pohang Earthquake. In the case of heterogeneous and/or unsteady tectonic stresses (the possibility of such a situation is indicated above by a non-stationarity of Σ ; for example, the Mw5.4 Gyeongju earthquake^{2, 10, 19} occurred on 12 September 2016 about 40 km south of the Pohang EGS, could contribute to such a heterogeneous and/or unsteady stresses situation), the real probability of the Pohang Earthquake could be even higher. A strong indication of a possibility of a runaway rupture can also be seen from a very high worst-case probability of an event of the stimulation-zone size (this is a largest event for which equation 1 can be rigorously applied based on seismicity parameters observed in the stimulation zone). For the Pohang EGS, such an event has a magnitude of the order of Mw4– Mw4.5 (the rupture size of the order 10^3 m and the stress drop in the range 1–5 MPa). The temporal behaviour of the probability of a stimulation-zone size event during the Pohang EGS stimulation activity is similar to that calculated for the Pohang Earthquake (Fig. 3e). However, the probability of such an event has reached 50%–80% (Fig. 3c,d the dark-red line). Such a high probability of a stimulation-zone size event was already achieved immediately after the induced Mw3.3 event (Fig. 3e). However, there were no earthquakes of the strength between Mw3.3 and Mw5.5. This is an indication that a similar event may have become the 2017 Pohang Earthquake with an unstable runaway rupture. Thus, during the stimulation operations the Mw2.3 event should be considered critical, and a safe stimulation strategy would be to keep the induced seismicity approximately below Mw2.0.”

[2] Implications for safe drilling and fluid injection

Operation of deep drilling with fluid injection without inducing/triggering damaging earthquake is very important now and will be even more important in the future. The analysis proposed in this paper can be done real time with the drilling operation and may lead to a renovation in the safe drilling, as mentioned clearly on Lines 261-265. The authors calculated a probability of the Pohang-size earthquake in the present paper ($M_{max} = 5.5$). However, I strongly feel that only this demonstration is weak as a recommendation for safe drilling operation. The stimulation-zone size earthquake is the limit of M_{max} for which its probability can be estimated rigorously from the seismicity parameters for the Pohang earthquake sequence. Also, the probability for this M_{max} can be taken as a probability for a triggered earthquake that can propagate beyond the stimulated zone. Such an event may turn out to be small, but it may grow to the size of the Pohang earthquake or even larger. Thus, avoiding earthquakes beyond the stimulated zone can be a useful guide for safe drilling operation at least for the size of the stimulated zone in Pohang (M_{max} has to be set up to an acceptable limit when the stimulated zone becomes very large). It will be very, very useful for future drilling to construct a new diagram similar to Figure 3 using M_{max} corresponding to the size of the stimulation zone, demonstrating what could have been safe drilling operation in Pohang avoiding triggered events. A point of interest is whether or not any drilling operation possible without inducing the stimulation-zone size earthquake after $M_{3.3}$ event. Result on Denver earthquake in Figure 4 are interesting and important, but it can be reported as a new and comprehensive paper elsewhere (information given in this paper is not enough!).

Response: In addition to the lines 234-253, we have introduced one more panel in Figure 3 (3e), where we plot the requested diagram and we have discussed it as described above. We have correspondingly changed the caption of this Figure. Also, we have expanded slightly our concluding paragraph with a corresponding recommendation of “monitoring the worst-case probability of a critically large stimulation-zone size event as a proxy of the runaway rupture probability” (lines 290-292).

Finally, we do consider our results on the Denver earthquakes as a necessary part of our paper. These results demonstrate applicability of our approach to other regions and to other types of induced earthquake (fluid disposals instead of EGS). We hope that our paper will motivate additional research on the Denver case study elsewhere (e.g., a relocation of Denver events – the information unknown to us).

[3] “Probability” of earthquake in Figure 3 and in the main text

A small comment on a term "probability". In Japan, a period is always specified in a probability of an earthquake (e.g., probability in the next 30 years). So I wondered what period is referred to the probability in Figure 3 of the present paper. Lines 324-325 in Methods gives "probability of triggered events with magnitudes $\geq M$ occurring in the time period from the beginning of fluid injection until injection time t ". "Injection time" is not specified (there were multiple injections) and it should be "time" on the horizontal axis of Figure 3. I thus consider the probability as an instantaneous probability at a give time. Please clarify this and adding a definition on "probability" in figure caption will be useful to readers.

Response: As it mentioned above, the definition of the worst-case exceedance probability has been given on lines 178-184 of the second revision of the manuscript. Corresponding improvements have been also introduced on lines 353-354, 375-376 and 381.

I understand that reviewers should not raise new issues that were not commented in the first review in re-review of a paper. The above comments [1] and [2] may appear to be new, but it came from my initial comment that the authors' method is applicable only to earthquakes up to the size of the stimulated zone. I hope that the authors take those as constructive to make a good paper even better. I suggest publication of the paper after some revisions. I look forward to seeing the paper in print whatever the final version is.

Response: Thank you indeed!

Sincerely yours,

Serge A. Shapiro, Kwang-Hee Kim and Jin-Han Ree

Corresponding author:

Serge A. Shapiro,
Earth Science Department, Freie Universität Berlin, 12249, Malteserstr. 74-100, Building D,
12249, Berlin, Germany, E-mail: shapiro@geophysik.fu-berlin.de, Mobile: + 49-1716495415